# Attributing decadal climate variability in coastal sea-level trends

Sam Royston[1], Rory J. Bingham[1], and Jonathan L. Bamber[1,2]

[1]School of Geographical Sciences, University of Bristol, UK
[2]AI4EO, Technical University Munich, Germany

**Correspondence:** Sam Royston (s.royston@bristol.ac.uk)

**Abstract.** Decadal sea-level variability masks longer-term changes due to natural and anthropogenic drivers in short duration records and increases uncertainty in trend and acceleration estimates. When making regional coastal management and adaptation decisions, it is important to understand the drivers of these changes to account for periods of reduced or enhanced sea-level change. The variance in decadal sea-level trends about the global-mean is quantified and mapped around the global coastlines of the Atlantic, Pacific and Indian Oceans, from historical CMIP6 runs and a high-resolution ocean model forced by reanalysis data. We reconstruct coastal, sea-level trends via linear relationships with climate mode and oceanographic indices. Using this approach, more than one-third of the variability in decadal sea-level trends can be explained by climate indices at 24.6% to 73.1% of grid cells located within 25 km of a coast in the Atlantic, Pacific and Indian Oceans. At 10.9% of the world's coastline, climate variability explains over two-thirds of the decadal sea-level trend. By investigating the steric, manometric and gravitational components of sea-level trend independently, it is apparent that much of the coastal ocean variability is dominated by the manometric signal, the consequence of the open-ocean steric signal propagating on to the continental shelf. Additionally, decadal variability in the gravitational, rotational and solid-Earth deformation (GRD) signal should not be ignored in the total. There are locations such as the Persian Gulf and African west coast where decadal sea-level variability is historically small, that are susceptible to future changes in hydrology and/or ice mass changes that drive intensified regional GRD sea-level change above the global-mean. The magnitude of variance explainable by climate modes quantified in this study infers an enhanced uncertainty on projections of short- to mid-term regional sea-level trend.

## 1 Introduction

Sea-level variability at the coast is driven by a variety of global- to local-scale factors. Understanding the drivers of variability due to decadal-scale, climate variability in historical and contemporary observations improves our understanding of sea-level change and enhances our ability to predict future, near-term sea-level change. By subtracting climate-driven sea level from observations, a more consistent global-mean sea level can be obtained from altimetry (Nerem et al., 2018) and tide gauge data (Frederikse et al., 2018). One aim is to elucidate anthropogenically-driven sea-level change from climate variability (Hamlington et al., 2014, 2019). When climate variability can be explained, by reducing both the magnitude and auto-regressive nature of variability in the signal, linear-trend and acceleration standard errors can be reduced. This has been successfully applied globally (e.g. Hamlington et al., 2013; Nerem et al., 2018; Hamlington et al., 2020c) and regionally, from climate variability dominated by atmosphere-ocean interactions (e.g. Zhang and Church, 2012; Pfeffer et al., 2018; Richter et al.,

2020; Hamlington et al., 2020b; Wang et al., 2021; Pfeffer et al., 2022) and intrinsic, oceanic variability in eddy-rich regions (Sérazin et al., 2016). Understanding the driving mechanisms behind local, decadal sea-level change could lead to improved short- to medium-term forecasts of coastal sea-level change.

Regional sea level is projected to vary by 30% from the global-mean, according to climate model evaluations (IPCC, 2019). Spatial patterns in sea-level variability are driven by intrinsic and climate variability and anthropogenic-forcing, and their inter-actions, resulting in intensified sea-level change by regions (as demonstrated by climate models with historical forcing; Fasullo and Nerem (2018)). In particular, changes to ocean heat content and surface winds drive changes in ocean circulation, in turn affecting the location of fronts and mixed layer or thermocline depth, which induce a sea-level change (Fasullo and Nerem,

2018; Fasullo et al., 2020; Peyser et al., 2016; Richter et al., 2020, e.g.). Also variations to the gravitational, rotational and deformational (GRD) equipotential redistribute the sea surface, following anthropogenically-forced mass redistribution such as dam impoundment, groundwater extraction, or anthropogenically-driven ice mass loss (Wada et al., 2017; Meyssignac et al., 2017; Frederikse et al., 2020b). The remainder of the regional variation we describe hereafter as atmospheric and/or oceanic 'internal' variability. Hereafter, we will use the terms "intrinsic variability" and "climate variability" to describe and differen-

tiate these sources of variability. By "intrinsic variability" we mean the oceanic variability that is not driven by atmospheric forcing but is internal. We use "climate variability" to refer to atmospheric-oceanic variability intrinsic to the climate system, not driven by anthropogenic forcing. Local sea-level trends greater than 10 mm y$^{-1}$ on 10-year timescales and over 1 mm y$^{-1}$ on 30-year timescales are observed in areas of the Pacific Ocean driven by climate variability; climate variability may poten-tially contribute centimetres of sea-level change over any given 10 year period, which local planners and stakeholders need to

account for (Hamlington et al., 2020a). This climate variability may affect the magnitude of regional sea-level trends calculated over durations up to 50 years (Carson et al., 2015, 2019). Sea-level variability related (linearly) to climate indices shows larger correlation coefficients at coastal locations in tide gauge data than in open ocean altimetry (Wang et al., 2021). Statistically sig-nificant relationships between the steric and manometric components of sea level and climate variability have been identified, from models, reanalyses and geodetic observations (Pfeffer et al., 2018, 2022). These works validate model and reanalyses

data sets for sea-level studies and identify regions of the global ocean with explainable, climate-driven, interannual to decadal variability, masking anthropogenically-driven changes and modifying coastal flood risk about the long-term trend on decadal periods.

Decadal variability in global-mean sea level is dominated by the El Niño Southern Oscillation (ENSO) and Pacific Decadal Oscillation (PDO) and their evolution in time (Hamlington et al., 2013; Nerem et al., 2018; Hamlington et al., 2020c). Because

these signals are large and affect a large proportion of the global-ocean area (being equatorial / tropical), they dominate the global signal. But other climate processes, described by other major indices, of course also affect local sea-level variability (Woodworth et al., 2019).

The Pacific Ocean decadal sea-level variability is dominated by the ENSO and PDO processes (e.g. Zhang and Church, 2012; Hamlington et al., 2019). Additionally, the Southern Annular Mode (SAM) and Indian Ocean Dipole (IOD) can also

be related to sea-level variability in the Pacific Ocean (e.g. Frankcombe et al., 2015). The IOD co-varies with ENSO on interannual timescales, via atmospheric teleconnections the IOD affects equatorial wind anomalies, and Pacific Ocean sea-

level anomalies may transit through the Indonesian Throughflow. The relationship is weaker at decadal time scales, but there remains a significant correlation (Nidheesh et al., 2019). The IOD dominates sea-level variability in the Indian Ocean (Nidheesh et al., 2019). In the North Atlantic Ocean and North Sea, variability can be related to the North Atlantic Oscillation (NAO) and East-Atlantic Pattern (e.g. Frederikse et al., 2018; Kleinherenbrink et al., 2016). We extend these analyses by focusing only on coastal sea level and we remove the global-mean sea level at each time step to investigate regional, spatial differences.

Although the spatial variability of regional, decadal-scale sea-level trends are dominated by the steric component (Richter et al., 2020), manometric sea-level changes dominate variability at the coast (Penduff et al., 2019; Llovel et al., 2018). When a steric-driven disturbance in the open-ocean sea surface height nears a coast, the density-driven change over a shallowing water column cannot fully match that in the open ocean and a pressure gradient develops in the sea surface. The geostrophic balance is maintained by redistribution of mass onto the shelf, such that sea-level change at the coast exhibits a predominantly manometric signal (Landerer et al., 2007; Yin et al., 2010; Bingham and Hughes, 2012; Penduff et al., 2019). We therefore also investigate the components of sea-level variability at the coast.

Here we use a 53-year run of a high resolution ocean model to quantify and characterise decadal-scale sea-level trend variability at the local, coastal scale (with the global-mean removed). A comparison is made with the CMIP6 historical run ensemble mean and spread. Climate model runs will not, in general, mimic the timing of internal atmosphere-ocean variability correctly but should capture much of its magnitude in the ensemble spread. Observed regional variability can be greater than coarse models suggest (Meyssignac et al., 2017; Carson et al., 2019), hence we compare the high-resolution run with the CMIP6 ensemble. These model runs are computationally expensive and we discuss the potential to use the relationship between sea-level variability and climate indices. We project climate mode indices on to the leading principal components (PCs) of an empirical orthogonal function (EOF) decomposition to reconstruct decadal, coastal sea-level trend associated with climate variability. With our focus at the coast, the reconstruction is applied to satellite altimetry and tide gauge observations and the variability of coastal sea-level trends discussed.

## 2 Method

Although ENSO variability dominates the spatial pattern of sea-level variability on decadal timescales, we wish to investigate if particular climate processes dominate sea-level trends, at the coast, over different regions, and for each component of sea-level change.

Climate and high-resolution ocean model runs are used to quantify the variability in decadal sea-level trends at the coast, from each component part. A reconstruction of coastal, decadal sea-level trends using only standard climate mode indices is attempted that can be easily replicated, by projecting climate mode indices onto principal components (PCs) of decadal, coastal sea-level trends. Variability of the coastal, decadal trends in sea-level components (derived from the high-resolution ocean model sea-level components plus GRD) is characterised by an empirical orthogonal function (EOF) analysis for each major ocean basin. The principal components (PC) of these sea-level component modes are correlated against climate mode indices to identify if a climate mode index co-varies with any PC (of each sea-level component and in each ocean basin). For each

95 sea-level PC in turn of diminishing variance explained (per component and basin), the climate index with maximum correlation is projected onto a PC by a linear regression, until each climate index is used or the correlation is not statistically significant. Thus, the decadal sea-level variability that can be associated with climate variability is reconstructed by one climate index and regression coefficient for each EOF-PC mode, with the sum over reconstructed PCs giving the total sea-level variability as

Firstly, the magnitude of variance in regional sea level and its trend are determined from ocean models. These variance are

100 calculated for total sea level and its component parts of manometric (model ocean bottom pressure) and steric contributions, plus the gravitation, rotation and solid-Earth deformation response (GRD) contribution, from the deviation from the global-mean at each time step (refer to Gregory et al. (2019) for terminology).

We use output from a high-resolution (nominally $1/12°$), eddy-resolving ocean model (NEMO) run over 58 years at monthly resolution from 1958–2015, in which the component steric and manometric signals sum to the sea-level signal (Marzocchi

105 et al., 2015; Moat et al., 2016). We use the later 53 years of data, allowing for 5 years' of spin-up. The global-mean sea level is removed at each time step, since we are primarily interested in the regional variability about the mean. The '*zos*' variable for sea level does not include atmospheric pressure effects and are therefore not included in this assessment. Thus the processed sea level quantifies the direction and variability of spatial patterns and intensification in sea level caused by observed atmospheric forcing plus intrinsic, oceanic variability, excluding the inverse barometer effect. It is acknowledged that this variability will

110 include influence from anthropogenic sources, because the high-resolution model is forced by reanalysis atmospheric data rather than natural forcing. The magnitude of variability is compared against the ensemble mean and spread of variance for the same time period from an ensemble of 43 historical forcing CMIP6 models.

The observed absolute sea-level signal as observed by altimetry includes the GRD component that is spatially varying. The barystatic, global-ocean-mean volume change due to solid-Earth deformation is ignored in this study because we remove the

115 global-mean at each time step. We add only the spatial geoid signal to the sea surface height (SSH) from the models (with a global-ocean-mean of zero).

To quantify how much of the sea-level variability can be described by a relationship with climate indices, where the impact of sea-level change is highest - at the coast - we investigate reconstructing sea level from climate indices over a decadal timescale. We investigated both low-pass filtered and rolling linear trend in time for each coastal grid cell time-series and

120 found the strongest relationship in the latter (not shown). We assume first-order auto-regressive (AR1) noise, appropriate for monthly sea-level time series (e.g. Bos et al., 2013; Dangendorf et al., 2014; Haigh et al., 2014), by a generalised least-squares regression that solves for the annual- and semi-annual periodics as well as the trend. We apply an EOF analysis to the rolling decadal trends from the modelled component parts and compare against decadal trends in key climate indices.

It is acknowledged that there are limitations in using EOF analysis and a linear regression to associate climate variability

125 with sea-level variability. Of course, the EOF method identifies the largest variance for its leading mode, and each mode is orthogonal from that. Therefore even when care is taken to deseason and detrend the coastal, sea-level component time series, variability from specific physical drivers may be distributed into several EOF-PC modes. However, by reducing the spatial dimensions using EOF analysis we limit computational effort and redundancy in the analysis because of spatial covariance, and produce a small data set of EOF patterns and loadings with just one set of regression coefficients each.

To focus where the impact of sea-level change is highest, and because the EOF analysis determines orthogonal bases from the first principal component (PC) with largest variability, we only apply the analysis to coastal regions. We define coastal by distance to the nearest coastline, selecting those model grid centres less than 25 km distance from one of the low resolution coastlines in the global self-consistent, hierarchical, high-resolution geography (GSHHG) database (following Penduff et al., 2019).

Our aim is to relate climate mode indices with the sea-level variability EOFs. A multi-variate linear regression analysis could be used. Linear regression analysis only finds the analytical least square error fit in the case of Gaussian data (no auto-correlation in the time series), and with independent explanatory variables. Some studies have reduced the impact of multicollinearity of the explanatory variables by low-pass and high-pass filtering correlated climate indices, giving new indices that represent short and long time scale processes (e.g. Zhang and Church, 2012; Wang et al., 2021). Alternatively the inflation

of regression coefficients due to multicollinearity of the explanatory variables may be reduced by the use of a penalty or regularisation term, i.e. a ridge regression. A Least Absolute Shrinkage and Selection Operator (LASSO) regression approach has been successfully applied to steric sea-level change (Pfeffer et al., 2018) and ocean mass change (Pfeffer et al., 2022) associated with climate variability, on inter-annual and longer timescales. This approach relies on the user determining an appropriate penalty term, typically by a cross-validation.

We take a simpler approach that does not have tunable parameters. For the reconstruction, we rank all leading EOFs and retain those that describe at least 5% of the sea-level trend variability for each component of sea-level change. For each leading PC in turn, a linear regression is applied to the climate index with the highest correlation coefficient, provided the correlation is significant (by $t$-test with the degrees of freedom reduced due to the auto-regressive nature of the signal; Emery and Thomson (2001)). This approach capitalises on the orthogonality of the sea-level trend PCs and ensures each climate mode index is only used once and only when significant. The reconstruction sums the climate index multiplied by the regression coefficient for

each climate index-PC pair until all climate modes are used and/or the correlation is not significant against any PC. However, as the EOF analysis may split the variability from a given physical process into more than one mode, it is likely that the relationship between a single climate index and single PC will underestimate the total variance caused by climate variability. Thus, the reconstructed climate-associated sea-level trends produced should be thought of as lower limit at each location of the trend-variance about the mean-trend.

Reconstructed trends are compared against the variability of running trends from the model, giving the variance explained by the climate index regression, calculated as the percentage ratio of trend variance at each grid cell, of the model-minus-reconstructed residual over the model rolling trends. The variance explained by these reconstructions of course varies by the adequacy of a simple, linear model, and the number of leading principal component modes used in the reconstruction.

To validate our method, an example period of satellite altimetry data from 2008–2018 is taken. We compare the reconstruction for the trend from 2008–2018 against observed sea level from satellite altimetry. The reconstruction for running trends centred on 1968–2011 are compared against tide gauge observations at arbitrary locations, demonstrating locations where the variance explained appears to be good. The decadal trend variability from tide gauge observations and the reduction in variability explained by all significant PCs for manometric, GRD and steric sea-level change combined and for each basin, using

the reconstructed sea-level rolling trend at the nearest model grid cell to each tide gauge location. The tide gauge relative sea level are corrected for glacial isostatic adjustment (GIA) but we do not correct for contemporary GRD-induced or other sources of contemporary vertical land movement (VLM) because of the limited number of tide gauges with co-located and benchmarked GNSS sites, instead removing the mean trend from tide gauge observed data. Rolling trends from observation data are treated identically to that from model data, a seasonal signal is solved for within the regression design matrix (an annual and a semi-annual periodic) and the noise is assumed to have an AR(1) characteristic.

## 3  Data

### 3.1  High-resolution 58-year ocean model run: NEMO

The total sea-level signal is partitioned into steric and manometric sea level from the NEMO ORCA0083-N006 model run, details of which can be found in Marzocchi et al. (2015); Moat et al. (2016). The model is applied to a high resolution ORCA tri-pole grid (nominally $1/12°$) so is eddy-resolving, it incorporates a sea-ice model, and is forced by the Drakkar Surface Forcing data set version 5.2 (Dussin et al., 2016), from 1958 to 2015 inclusive. This data set derives ocean model forcing variables from the ERA40 and ERA-interim atmospheric data sets, and includes freshwater fluxes (precipitation and snow). Freshwater runoff is added as seasonal cycles, and does not exhibit interannual changes. Because of deficiencies in the freshwater forcing, a moderate relaxation of surface salinity to climatology is applied and a freshwater budget restoration is applied at time steps when a deficit is found (and only applied to areas with precipitation).

Steric sea level is calculated using the TEOS-10 equation of state (TEOS-10, 2008) applied to temperature and salinity modelled values at each model depth level. The total and steric sea-level anomaly are affected by the Boussinesq approximation in this model setup (Greatbatch, 1994; Griffies and Greatbatch, 2012). This effect and the global-ocean-mean atmospheric pressure effect are removed by subtracting the global-mean of each sea-level anomaly at each time step.

The manometric sea-level component is taken to be equal to the ocean bottom pressure anomaly, converted from pressure to mm change in height.

The ocean model has been shown to match observed variability well (Marzocchi et al., 2015). We additionally check that the linear trend from last decade of the model run, 2005-2015, with GRD added matches the altimetry observation of absolute sea-level trend (Supplementary Information Fig. S1). In this analysis, only coastal locations, within 25 km of the low resolution GSHHG coastline, are considered.

### 3.2  CMIP6 climate model historical runs

We ensure that sea-level trend variance given by the NEMO model is appropriate for our aim, by checking the variance magnitude lies within the envelope of sea-level trend variance from historical-forced model runs from the 6[th] Climate Model Intercomparison Program (CMIP6; Eyring et al. (2016)).

Model run data was obtained via JASMIN (UK data and storage facility, https://jasmin.ac.uk/users/access/) and may also be obtained from the WCRP portal (ESGF, 2021). Historical forcing (esm-hist) has been run in CMIP6 from 1850 to 2014 inclusive. Here we take the 'zos' variable sea surface height, monthly means from January 1958 until the end of the run in December 2014 (noting this is one year shorter than the high-resolution NEMO run). The *zos* variable is interpolated onto a regular lat-lon $1/4°$ grid for each run and the global-ocean mean is removed from each time step of each model. Appendix A lists the 43 CMIP6 model setups analysed in this study. In this analysis, only coastal locations, within 25 km of the low resolution GSHHG coastline, are considered.

### 3.3 Climate mode and oceanographic indices

Major climate variability is represented by indices, derived from various atmospheric and oceanic observables, such as air pressure at sea level, sea surface temperature and surface wind speed or its gradient. Here, we determine the correlation of the principal component time series of rolling sea-level trends with the rolling trends of 6 major climate includes: the Pacific Decadal Oscillation (PDO; NOAA-NCEI (2020c)), El Nino Southern Oscillation (ENSO; Multivariate ENSO Index, NOAA-NCEI (2020a)), North Atlantic Oscillation (NAO; NOAA-NCEI (2020b)), Arctic Oscillation (AO; NOAA-CPC (2020)), Southern Annular Mode (SAM; Marshall (2020)) and Indian Ocean Dipole (IOD; GCOS (2020)).

Additionally, the effect of the Atlantic Meridional Overturning Circulation (AMOC) is investigated. The AMOC index is calculated here from the NEMO model runs, as an anomaly at each time step. The index is computed as the principle component of the low-pass filtered (1-year running mean) and zonally-integrated meridional transport, in Sv, and then the rolling trend is calculated from this index.

It is acknowledged that these indices are not independent of each other. However the EOF patternd and principal components of coastal sea-level change are orthogonal. Therefore only one climate index is associated with each PC and not repeated in the reconstruction.

### 3.4 Absolute sea level: Satellite altimetry

Absolute sea level is defined from the ESA SLCCI v2 multi-mission, gridded product on a 1/4° grid with the most up-to-date corrections and processing available (Legeais et al. (2018); ESA (2018)). The standard global-mean trend GIA correction is applied at each grid point (-0.21 mm y$^{-1}$ for the ICE6G-VM_D GIA model (Peltier et al., 2015; Peltier, 2018)). This global-mean value represents the shift in geopotential surface of the geoid (Tamisiea, 2011). Because we are interested in the spatial distribution of sea-level trends, the spatially redistribution of the geoid by GIA is also applied (Tamisiea, 2011), as a trend correction derived from the spherical harmonic coefficients provided by Peltier (2018). Contemporary GRD variability driven my mass redistribution affects the sea surface. Satellite altimetry observes spatial changes to the geoid from a centre of mass and should be corrected for the global-mean volume change due to solid-Earth deformation (Frederikse et al., 2017). We correct the gridded satellite altimetry for solid-Earth deformation associated with recent mass loading of the oceans, following Frederikse et al. (2020b) and using their published data (Frederikse et al., 2020a).

## 3.5 Tide gauge observations

Tide gauge observations are obtained from the Permanent Service for Mean Sea Level (Holgate et al., 2013) for revised local reference (RLR) stations only. The relative sea level is corrected for GIA (Peltier et al., 2015; Peltier, 2018) only, and no account is made for contemporary GRD-induced or other VLM. Because we are primarily interested in the temporal variability of sea-level trends associated with climate variability, in the Figures we remove the time-mean trend; the linear trend correction does not affect the results. Non-linear solid-Earth deformation from GRD and other sources such as ground compaction and building load are not accounted for and will present in the tide gauge trend variability. Monthly mean time series, omitting flagged data, are used to determine rolling trends and periods where less than 50% of data are missing in each rolling decade are omitted.

## 3.6 Gravitation, rotation and solid Earth deformation changes

The ocean models do not include any GRD changes. Observations by their nature include GRD effects. The solid-Earth deformation changes modify the basin-shape and therefore global volume. For absolute sea level observed by satellite altimetry the global-ocean mean solid-Earth deformation from contemporary mass redistribution and global-mean glacial isostatic adjustment (GIA) effects are usually subtracted as a correction. Altimetry observes regional redistribution of the geoid when the anomalies are taken from a mean sea surface. Relative sea level observed by tide gauges include solid-Earth deformation. To compare model data with observations, we add variability from a sea-level fingerprint method applied to comprehensive data sets of land and cryospheric mass loading. The data set has been used to estimate the GRD effect on global-mean and basin-mean sea-level trends where the time-varying vertical land movement was used to correct tide gauge relative sea-level records (Frederikse et al., 2020b, a). The GRD geoid fingerprints include the effect of mass changes in glaciers, the Greenland and Antarctic Ice Sheets, including uncharted glaciers and peripheral glaciers, and from natural terrestrial water storage (TWS), dam retention (or reservoir impoundment) and groundwater depletion. Here, we determine decadal rolling trends from the geoid variability of the sea-level fingerprint since 1958 and ignore the barystatic (global-mean) component since we are interested in the spatial variability.

## 4 Results and Discussion

### 4.1 Variance of decadal sea-level trend

Over the 58-year NEMO model run, coastal sea-level rolling trends vary in time by a mean standard deviation of 3.6 mm $y^{-1}$, with some locations displaying a standard deviation in trends over 7.5 mm $y^{-1}$ (Fig. 1), which is significantly larger then the mean trend over the modelled period of 2.2 mm $y^{-1}$.

The variance of total SSH in the NEMO model run generally sits within the spread of variance in the CMIP6 ensemble. There is increased variance in the CMIP6 ensemble mean compared with the NEMO model run in the Northern Hemisphere, particularly in the Beaufort Straights, Hudson Bay, and the North Sea into the Baltic Sea, and to a lesser extent in the Mediterranean

and Black Seas (comparing Fig. 1a and 1b). It is well known that many coarse-resolution climate models do not reproduce sea level in semi-enclosed seas well, because of resolution limits on fluxes with the ocean basins (Adloff et al., 2018; Meyssignac et al., 2017). In contrast the NEMO model run displays a larger variance in sea-level trends around the coast of Greenland, in the tropical West Pacific and west coast of Australia, in the Caribbean Sea and around Chesapeake Bay. However most of these differences lie within the CMIP6 ensemble spread and are therefore not significant.

This analysis confirms there is high variability in decadal sea-level trend in the Western Tropical Pacific Ocean. On the eastern side where the coast faces the open ocean, the signal is dominated by steric changes, but through the Indonesian Throughflow and in the marginal seas, the signal becomes manometric in nature (Fig. 2a,b). For most of the coastal locations, the variance in sea-level trend is dominated by manometric sea-level change; in 48.2% of coastal locations defined from the NEMO grid the manometric sea-level change shows the largest contribution to variance, in 32.3% of locations steric sea level is dominant and in 17.5% of locations the GRD effect is dominant.

Around the Greenland coast, in the vicinity of major ice mass loss that is variable in time, there is a large variability in decadal trends due to the GRD effect (Fig. 2c). The GRD signal driven by glacier and ice sheet mass loss also contributes more than one-quarter to the variability around south Alaska, Hudson Bay and the Canadian Arctic and Iceland, and to a lesser extent around the Patagonian ice sheet. It is noted that this analysis relates to absolute sea level equivalent, and VLM contributing to relative sea level is not considered. There is also a notable contribution from GRD in areas around the major river basins or other hydrology impacts (like groundwater abstraction or dam retention) where interannual variability in TWS is large, for example around the Amazon River basin, Niger River basin and the Persian Gulf.

Even though in many of those regions the variability in decadal trend around the global-mean is small (around 1 mm y$^{-1}$), where the dominant contributions are GRD and/or local steric contributions from hydrology, future changes may exacerbate the trend. For example, in the Persian Gulf, decadal sea-level trends are dominantly affected by GRD changes over direct oceanographic changes in addition to global-mean sea-level rise. It is noted here there is a complex GRD signal from both hydrology and glacier mass loss. On the south-east African coast, as in much of the South Atlantic, where there are very few long-duration tide gauge measurements, the trend variability is small (less than 1 mm y$^{-1}$) in both the CMIP6 ensemble mean and NEMO model (Fig. 1a,b). Here, sea-level variability is strongly influenced by GRD effects from the variability of hydrology, with the Amazon, Niger, Congo and Zambezi/Okavango basins driving more than 40% of sea-level trend variability in places (Fig. 2c). Long-term anthropogenic or climate change impacts on the hydrology in these locations are likely to intensify the regional trend about the global-mean.

## 4.2 Climate index reconstruction of sea-level trends

We reconstruct decadal running sea-level trends from climate index trends by ocean basin for steric and manometric sea level separately, and then combine the reconstructions. When compared with the NEMO model sea-level rolling trends at each coastal grid point (from which the regression coefficients were derived), the reconstruction displays statistically significant correlations ($r > 0.32$ for a two-sided $t$-test at 95% CI with an auto-correlation of 0.5 in the rolling trends) along much of the global coastal locations (Fig. 3a). There is a notably poor correlation around South-East Africa, where the interaction of

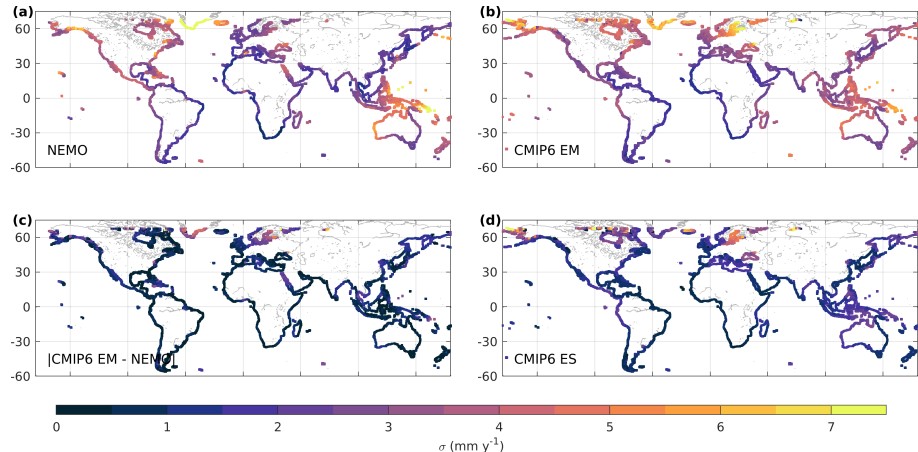

**Figure 1.** Comparison of the variability in decadal, sea-level trends between NEMO and an ensemble of CMIP6 model runs. Standard deviation of decadal trends (mm y$^{-1}$) in sea surface height from the NEMO model (a), the ensemble mean EM of the CMIP6 model runs (b), the absolute difference between NEMO and CMIP6 EM (c), which may be compared against the CMIP6 ensemble spread, ES (d). The SSH from the NEMO model and ensemble mean of CMIP6 model runs (a,b) include GRD.

the Benguela and Agulhas currents and upwelling may interrupt far-field climate-driven sea-level variability. The proportion of sea-level trend variance reconstructed by climate variability through our approach is small around much of the global coast (oranges, greys, pinks in Fig. 3). The reconstruction explains more than half of the decadal sea-level variance along
the American continent's west coast and in the Tropical Pacific Ocean and Indonesian Throughflow to the West Australian coast and west coast of Asia (blues in Fig. 3b). The primary mode explains more than half of the decadal trend variance in the semi-enclosed seas of East Mediterranean, Black and Baltic Seas. Along the traditionally under-sampled West African coastline between 10°N and the Mediterranean outflow at Gibraltar Straits and between 5°S and 30°S, the reconstruction explains between one-third and one-half of variability in decadal trends. The approach explains more than half of decadal trend
variance over 25.8% of the global, non-polar coastal ocean, and with greater success in the Pacific Ocean where the ENSO variability is dominant (Table 1). Table 1 presents the proportion of grid cells located in each basin where the reconstruction explains more than one-third, one-half or two-thirds of the decadal sea-level trend variance (not area-averaged). The column 'SSH' refers to a reconstruction using only total SSH EOFs and the column 'DSL + steric' refers to a reconstruction summed from the EOFs of all sea-level components. For each major basin, the approach can explain more than one-third of the decadal
trend variance for 24.6% to 73.1% of coastal locations (Table 1).

When considering the proportion of variance explained for a coastal location, the manometric sea-level signal becomes important. The first principal component modes from manometric and steric sea level are very similar to that from sea surface height and have highest correlation with the same climate indices, colored except for the influence of AMOC in the Atlantic Ocean (Supplementary Information Table S1 and Figs. S5 and S11). However, by adding the contribution from each component
separately there is a marginal improvement in the overall variance explained by this approach (Table 1). Splitting the variance

| Coastal Domain | > Var. Exp. | SSH | DSL + steric |
|---|---|---|---|
| All basins | one-quarter | 58.9 | 54.9 |
| | one-third | 48.2 | 44.5 |
| | one-half | 29.3 | 25.8 |
| | two-thirds | 7.2 | 12.3 |
| | three-quarters | 1.8 | 5.7 |
| Atlantic | one-quarter | 52.4 | 40.8 |
| | one-third | 41.2 | 28.7 |
| | one-half | 21.4 | 8.7 |
| | two-thirds | 4.8 | 0.0 |
| | three-quarters | 0.1 | 0.0 |
| Pacific | one-quarter | 78.9 | 82.0 |
| | one-third | 66.9 | 73.1 |
| | one-half | 46.6 | 54.2 |
| | two-thirds | 12.7 | 32.8 |
| | three-quarters | 4.6 | 15.3 |
| Indian | one-quarter | 24.4 | 29.9 |
| | one-third | 19.6 | 24.6 |
| | one-half | 8.7 | 9.1 |
| | two-thirds | 0.3 | 0.0 |
| | three-quarters | 0.0 | 0.0 |

**Table 1.** The proportion of coastal grid cells defined from the NEMO grid with more than one-quarter (25%), one-third (33%), one-half (50%), two-thirds (67%) and three-quarters (75%) of decadal trend variance explained by the reconstruction. The values shown here are from assessment of the 'coastal domain' and the NEMO model runs. 'SSH' refers to the reconstruction using EOF and PC modes for the full SSH signal and 'DSL + steric' refers to the reconstruction by components, for manometric dynamic sea level, GRD and steric sea level associated with climate indices separately and then summed.

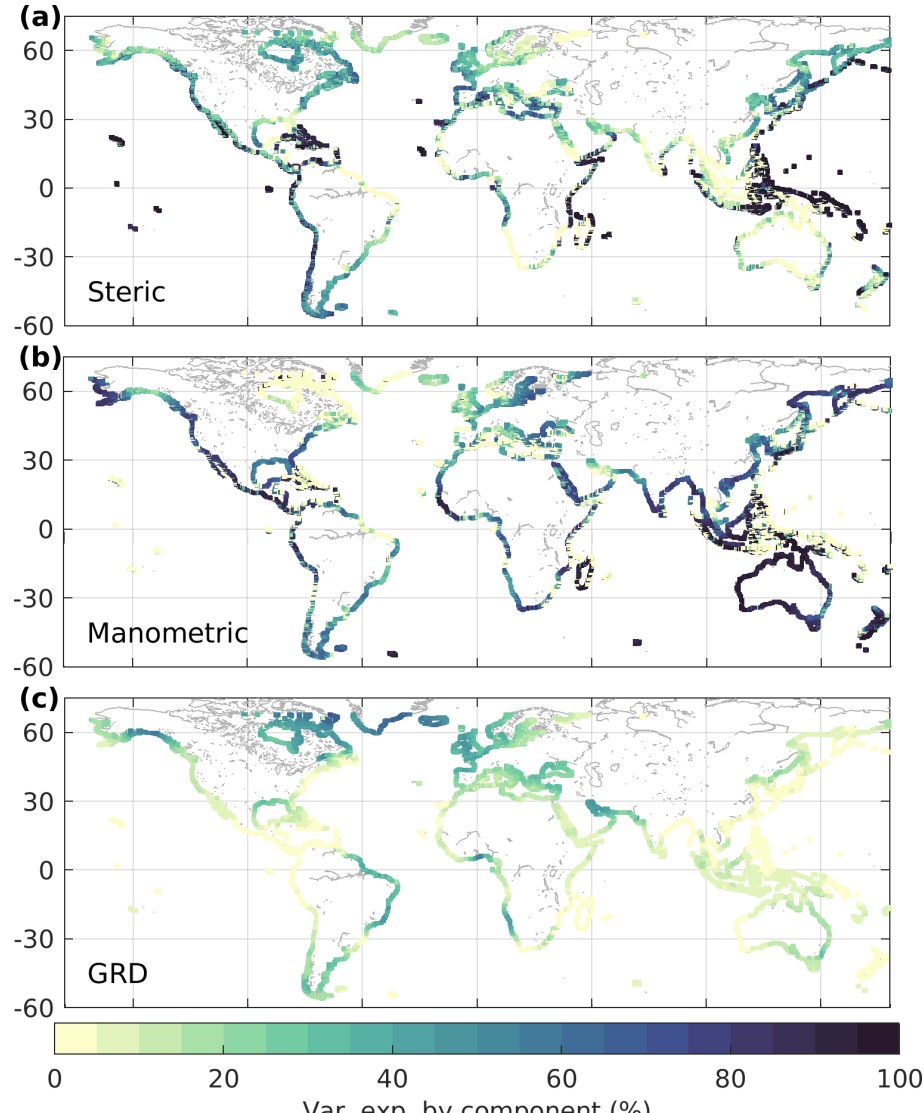

**Figure 2.** Proportion of variance explained (%) by sea-level components of the rolling decadal trends in sea surface height from the NEMO model: by steric sea level (a), manometric sea level (b) and GRD respectively (c). Fig. 1a presents the magnitude of the variability (standard deviation).

into component parts allows the EOF analysis to determine more specific spatial patterns for each component part, whereas the total SSH is dominated by different physical processes in different areas (Fig. 2).

Oscillations in observed tide gauge decadal trends are explained well by the climate index reconstruction in some regions, in particular across the tropical Pacific and the coast of the Americas and on the Atlantic West coast (examples are given in Fig. 4). Where the majority of the signal is manometric, for example the West coast of Australia and the Gulf of Maine

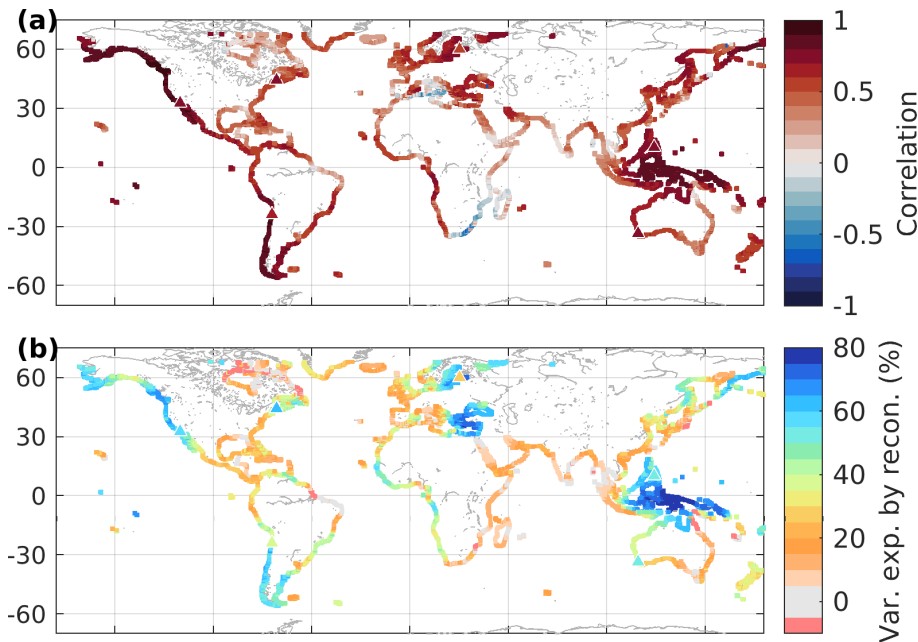

**Figure 3.** Comparison of decadal, rolling sea-level trends from the NEMO model plus GRD, and reconstruction using only climate indices: Pearson's correlation coefficient (a) and variance explained (b; %). Results here sum reconstructed decadal trend from manometric sea level plus steric sea level by ocean basin. Triangles denote the locations of tide gauge observations shown in Fig. 4.

(Fig. 4c,e), coastally-trapped waves propagate along the continental slope and shelf. Where the continental shelf is narrow, the reconstructed sea level is predominantly steric (Fig. 4b). In the tropical West Pacific, the dominant ENSO steric signal directly impacts tropical West Pacific tide gauge sites on the oceanward (eastern) coast, but the signal propagates through the Indonesian throughflow and around the island as a manometric signal, so that by Cebu the signal is predominantly manometric (Fig. 4a). The tide gauge data do not have contemporary VLM removed (except GIA), nor the nodal tide or inverse barometer correction made. For visualisation we simply remove the mean trend for the whole period considered.

For those locations with large magnitude variability in the trend, i.e. with a standard deviation larger than the global-mean trend of 3.5 mm y$^{-1}$ (oranges and yellow is Fig. 1a,b), typically more than half of that decadal signal can be attributed to climate forcing and reconstructed from climate indices. For 10% of the coastal locations in this study, over two-thirds of the regional decadal sea-level trend about the global-mean can be quantified by a linear relationship with climate index data (Table 1).

### 4.3 Climate effect on recent coastal sea-level trends

The reconstructed sea-level variability due to primary climate modes can be compared against the spatially-comprehensive satellite altimetry data, with the global-mean trend removed (to emphasise regional patterns in sea-level change).

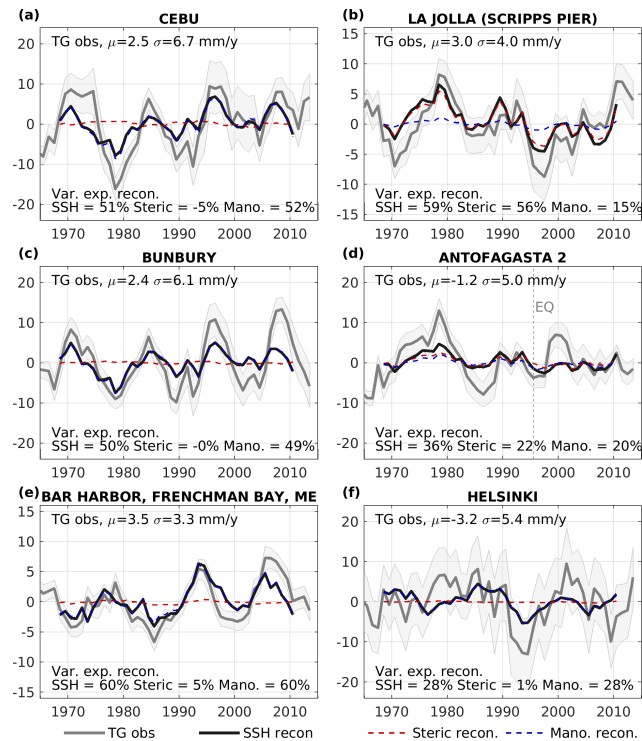

**Figure 4.** Observed sea-level trends in tide gauge observed sea level (mm y$^{-1}$; grey solid lines, light grey shading presents $1\sigma$ trend error estimates, triangles on Fig. 3), and the reconstructed decadal trends from all PCs for the appropriate basin, for steric sea level (red dashed), manometric plus GRD sea level (blue dashed) and the sum (black solid). Note the time-mean sea-level trend is removed from each tide gauge observed data for visualisation.

For a recent decade of observations, 2008–2018 inclusive, the reconstruction of sea-level trend anomaly along the coast associated with climate indices (Fig. 5) captures the dipole of sea-level trend anomaly across the Pacific Ocean and at least one-third of the decadal trend anomaly in the Caribbean Sea and Black Sea, and the sign of the trend anomaly in south Greenland, the Baltic Sea and North West African coast. The reconstructed signal is not as strong as observed. In some regions, such as the Gulf of Mexico, the reconstructed trend associated with climate variability (Fig. 5b) displays the opposite sign to the observed trend (Fig. 5a); in these locations in this period, the observed minus reconstructed trends have a larger magnitude trend from the global-mean. The histogram of trend anomalies for this period are markedly more Gaussian when reconstructed with climate-index-related variability removed than the unaltered observations.

Notably in all basins, by removing the reconstructed variance by climate indices, the mean (median) coastal sea-level trend for 2008–2018 is increased, by 0.7 (0.2) mm y$^{-1}$ globally (Table 2). It is noted that we expect climate variability has affected the global-mean sea level over the same period which we do not account for here, but we may conclude that coastal sea levels have been suppressed by the phase of climate variability in 2008–2018, compared with the entire-ocean mean.

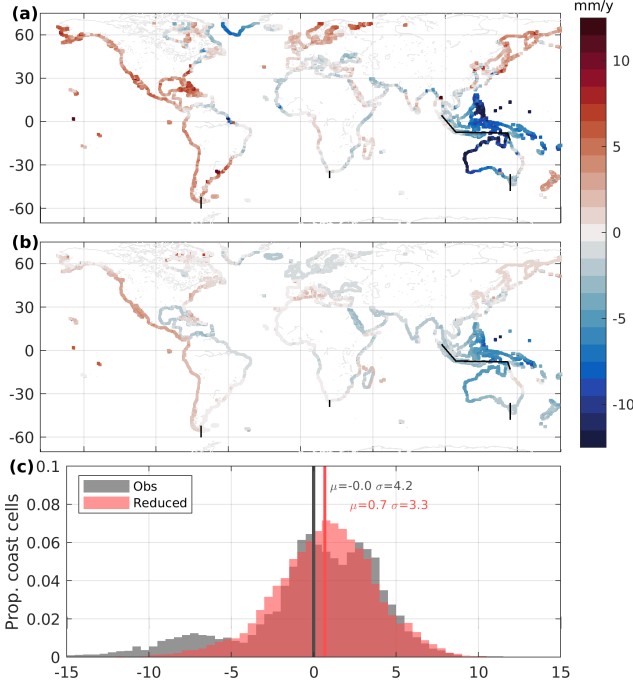

**Figure 5.** Comparison of 2008–2018 trend anomaly (mm y$^{-1}$) from satellite altimetry and reconstruction. Observed decadal trend in satellite altimetry with the global-mean removed (a) and the reconstructed decadal trend for each basin and component combined (b), with histogram (c) for all coastal grid cell locations for observations from satellite altimetry (grey) and the altimetry minus the reconstruction (so-called 'reduced' trends, red).

| Statistic | Coastal Domain | Obs. | Reduced | Diff. |
|---|---|---|---|---|
| Std. dev. | Global | 4.2 | 3.3 | -1.0 |
| | Atlantic | 2.8 | 3.0 | 0.2 |
| | Pacific | 5.2 | 3.3 | -1.9 |
| | Indian | 4.6 | 3.7 | -1.0 |
| Mean | Global | 0.0 | 0.7 | 0.7 |
| | Atlantic | 0.8 | 1.1 | 0.3 |
| | Pacific | -0.1 | 0.6 | 0.7 |
| | Indian | -3.0 | -1.0 | 2.0 |

**Table 2.** Statistics of the global-coastal sea-level trend from 2008–2018, observed by satellite altimetry and when reduced by the reconstructed sea level expected from climate index reconstruction for each basin (mm y$^{-1}$).

|               | Atlantic |       | Pacific |       | Indian  |       |
| Component     | Index    | beta  | Index   | beta  | Index   | beta  |
| ------------- | -------- | ----- | ------- | ----- | ------- | ----- |
| Steric PC1    | AMOC     | -0.83 | ENSO    | 1.60  | ENSO    | -0.95 |
| Manometric PC1| AMOC     | -1.01 | ENSO    | 1.64  | ENSO    | -1.24 |
| GRD PC1       | AMOC     | -0.67 | AMOC    | -0.88 | AMOC    | -0.84 |

**Table 3.** The linear trend coefficient between decadal trend in climate index and the first PC time series in each basin and for each omponent of sea level.

## 4.4 Sources of decadal variability and caveats

Interannual sea-level variability can develop purely as a response to non-linear interactions in oceanic intrinsic variability, and can evolve from seasonal forcing as strongly as from atmospheric forcing (Llovel et al., 2018). Oceanic intrinsic variability
exceeds the forced response to atmospheric forcing at some lengthscales over several years in high resolution ensembles (Sérazin et al., 2015, 2016; Llovel et al., 2018; Penduff et al., 2019). Thus, sea-level variability is the aggregated response, over integrating time-scales, of both atmospheric forcing and intrinsic variability in the system. It is acknowledged that the approach could be improved by comparing the PCs with phase-lagged trends in the climate indices and/or with other metrics describing the forcing.

Globally, the ENSO and PDO have been shown to dominate the decadal-scale variability of coastal sea level over other climate processes (e.g. Hamlington et al., 2013; Nerem et al., 2018). Because the PDO or ENSO decadal variability dominate the power in sea-level variability, the EOF bases on a global data set are forced to be orthogonal to that mode. To investigate other drivers, we further mask the data into three oceanic basins, the Atlantic, Pacific and Indian Oceans. By focusing on each oceanic basin in turn, the dominant mode(s) from each region can be identified (Table 3; Supplementary Information Table S1
and Figs S4 to S18). Since we aim to reconstruct decadal-scale sea-level trends, it is difficult to make direct comparisons with previous studies of climate variability and sea-level change. The dominant influence of ENSO and PDO indices with Pacific and Indian Ocean sea level agrees with other works (e.g. Zhang and Church, 2012; Hamlington et al., 2019; Wang et al., 2021; Pfeffer et al., 2018, 2022). In the North Atlantic Ocean, our approach finds a stronger relationship between sea-level variability and AMOC, followed by the AO. Previous studies that include interannual variability find strong relationships with the NAO
(Frederikse et al., 2018). Studies that separate the sea-level components have found steric sea-level variability strongly relates to the Atlantic Multidecadal Oscillation (AMO) (Pfeffer et al., 2018) and our approach results in a similar relationship between manometric sea-level change and the AO as Pfeffer et al. (2022) (Supplementary Information Fig. S12).

Generally climate models display lower sea-level variability than observed (Carson et al., 2015). In particular the CMIP5 models were found to simulate sea-level variability comparable to observations but showed bias in trends (Meyssignac et al.,
2017); other authors found that the long term memory, power-law character of sea level in CMIP5 models is too small, indicating the sea-level variability is too short-lived (Becker et al., 2016). Here, the use of a high resolution model goes some way

to minimise that influence but we caution that the resulting reconstructed sea-level variability and its trend should be thought of as a minimum rise or fall in sea level expected with climate index evolution.

It is acknowledged that the EOF patterns and their PCs developed here are somewhat model-dependant and because of the linear approach taken may incorporate anthropogenic as well as internal forcing patterns. Because of the multicollinearity in the explanatory variables (climate mode indices) and auto-correlation in the variables, there are limitations on any type of regression analysis that attempts to associate climate variability with sea-level variability. The EOF analysis may split the variability from a given physical process into more than one mode, weakening the relationship with any single climate mode index. Therefore, the reconstructed climate-associated sea-level trends produced should be thought of as lower limits at each location of the trend-variance about the mean-trend. A multi-variate approach with regularisation could be applied instead.

## 5    Conclusions

The current temporal duration of high quality sea-level data with good spatial coverage conflates with the typical auto-correlated, integrated, long-memory time scale of variability in major atmosphere-ocean climate modes, recently shown for steric sea level variability particularly in the Atlantic by Pfeffer et al. (2018) and for the open Pacific Ocean by Hamlington et al. (2020a). Therefore much of the current linear trend in steric and manometric components of sea level can be reconstructed from climate index data in some parts of the global-coast. This enables the possibility that observed sea level and its components can be reduced for climate variability, as has been applied for the total sea-level signal at the global-scale (Nerem et al., 2018; Hamlington et al., 2020c) and spatial distribution in the Pacific Ocean (Hamlington et al., 2020b, a) and globally for time series (Wang et al., 2021).

We present an analysis of the variance in local, short-term (decadal) sea-level trends about the global-mean around the Atlantic, Pacific and Indian Ocean coastlines. These data are an indicative lower-bound of uncertainty in regional short-term trend deviations from global-mean projections. The standard deviation of decadal trend exceeds the global-mean of $3.5\ \mathrm{mm\,y^{-1}}$ along the eastern North Pacific, western tropical Pacific, New Zealand and western Australian coastlines, eastern Indian Ocean, parts of the Caribbean Sea coast and western Atlantic coastline including Greenland coast, and many semi-enclosed seas.

For a recent decade of observations, from 2008–2018, the global-coastal mean sea level (here defined within 25 km of the coast and ignoring the Arctic and Antarctic coastlines) has been suppressed by climate variance, by $0.7\ \mathrm{mm\,y^{-1}}$ in the coastal-mean. In particular, this increase is greatest in the Indian Ocean basin ($2.0\ \mathrm{mm\,y^{-1}}$ greater).

More than half of the decadal sea-level trend can be explained by a linear regression with major climate index trends at around 25% of global coastal (within 25 km of the coast) locations, rising to 54% of grid cells around the Pacific Ocean. The ENSO and PDO variability dominates here, and the open ocean variability observed by many previous studies extends to and around the coast most notably in the West Tropical Pacific and along the coast of the Americas. Our approach has no lag or lead time introducedand explains less than one-third of the decadal variance in the low latitude East Pacific Ocean and in the mid-latitudes of the West Pacific. In the Indian Ocean, our method is most successful in the eastern basin, where the propagation of ENSO-related sea-level disturbance dominates through the Indonesian Throughflow and therefore dominates

the first EOF mode, explaining more than 40% of decadal variance along the West Australia coast, but less than 20% elsewhere. In the Atlantic Ocean our approach works well in the Baltic, Black and Eastern Mediterranean Seas and along the west coast of North Africa (Eastern Tropical Atlantic Ocean) with more than 50% variance explained in places, but is less informative on the north eastern Atlantic margin. Notably this region of North Africa, and other regions where the variance explained is lower but still statistically significant such as the Caribbean Seas and Bay of Bengal, have a lack of good quality and long duration tide gauge data by which to evaluate decadal-scale variability needed to make helpful forecasts of sea-level trends over the mid-term. The dominant influence of ENSO and PDO on sea-level change in the Pacific and Indian Oceans and the influence of AO on Atlantic Ocean manometric sea-level change match previous studies (Zhang and Church, 2012; Hamlington et al., 2019; Wang et al., 2021; Pfeffer et al., 2022). Our approach finds a strong relationship between AMOC and decadal sea-level change in all basins.

The variability of GRD in total sea-level trend should not be ignored over timescales of the order of 10 years (Fig. 2). The variability in decadal-scale coastal sea-level trend over much of the coastal ocean is dominated by manometric and GRD sea-level components rather than steric sea level. The coasts where steric sea-level trend variability dominates the signal are mostly tropical or low latitude towards the west of ocean basins and at the oceanic extent of the continental shelf. Sea-level disturbances that originate as steric in the open ocean propagate onto the continental shelf as a mass signal at the local scale. Thus, sea-level trends in the open ocean that can be associated with steric forcing need to be propagated accordingly onto the shelf, i.e. using high-resolution models, to adequately forecast variability at the coast. Future anthropogenic or climate change influences on hydrology and ice mass change driven GRD will disproportionately affect some regions that historically display low decadal variance, such as the Amazon Basin, the west coast of Africa from Niger, Congo and Zambezi hydrology, and the Persian Gulf.

**Appendix A: CMIP6 models**

The CMIP6 models used in this study are given in Table A1.

*Data availability.* The CMIP6 model run and NEMO model run outputs are available to download from their original sources (ESGF (2021); www.jasmin.ac.uk). Additionally, CMIP6 model runs are available from the WCRP data portal at https://esgf-index1.ceda.ac.uk/ search/cmip6-ceda/. Public archives of the NEMO ORCA0083-N006 model run are found at http://gws-access.ceda.ac.uk/public/nemo/runs/ ORCA0083-N06/means/ (Coward, 2016). The NEMO ocean model code and its documentation are available from https://www.nemo-ocean. eu. We use SSH data from satellite altimetry from the ESA SLCCI v2 project (ESA, 2018); GRD data provided by Frederikse (Frederikse et al., 2020a); and climate mode indices as cited in the text.

The data produced in this analysis and used to create the Figures and Tables is available to download from Zenodo with DOI: 10.5281/zenodo.5849268 at https://zenodo.org/record/5849268#.YeFtDGjP2Uk.

| Model Name | Model Centre | Resolution |
| --- | --- | --- |
| ACCESS-CM2 | CSIRO-ARCCSS | - |
| ACCESS-ESM1-5 | CSIRO | - |
| BCC-CSM2 | BCC | MR |
| CAMS-CSM1-0 | CAMS | - |
| CanESM5 | CCCma | - |
| CAS-ESM2-0 | CAS | - |
| CESM2 | NCAR | - |
| CESM2-FV2 | NCAR | - |
| CESM2-WACCM | NCAR | - |
| CESM2-WACCM-FV2 | NCAR | - |
| CIESM | THU | - |
| CMCC-CM2 | CMCC | HR4 |
| CMCC-CM2 | CMCC | SR5 |
| E3SM-1-0 | USDOE | - |
| E3SM-1-1 | USDOE | - |
| E3SM-1-1-ECA | USDOE | - |
| EC-Earth3 | EC-Earth-Consortium | - |
| EC-Earth3-AerChem | EC-Earth-Consortium | - |
| EC-Earth3-Veg | EC-Earth-Consortium | - |
| EC-Earth3-Veg | EC-Earth-Consortium | LR |
| FGOALS-f3-L | CAS | - |
| FGOALS-g3 | CAS | - |
| FIO-ESM-2-0 | FIO | - |
| GFDL-CM4 | GFDL | - |
| GFDL-ESM4 | GFDL | - |
| GISS-E2-1-G | NASA-GISS | - |
| GISS-E2-1-G-CC | NASA-GISS | - |
| GISS-E2-1-H | NASA-GISS | - |
| INM-CM4-8 | INM | - |
| INM-CM5-0 | INM | - |
| IPSL-CM6A | IPSL | LR |
| KIOST-ESM | KIOST | - |
| MIROC6 | MIROC | - |
| MPI-ESM-1-2-HAM | HAMMOZ-Consortium | - |
| MPI-ESM1-2 | MPI-M | LR |
| MPI-ESM1-2 | MPI-M DWD DKRZ | HR |
| MRI-ESM2-0 | MRI | - |
| NESM3 | NUIST | - |
| NorCPM1 | NCC | - |
| NorESM2 | NCC | LM |
| NorESM2 | NCC | MM |
| SAM0-UNICON | SNU | - |
| TaiESM1 | AST | - |

**Table A1.** List of CMIP6 models used in this study.

*Author contributions.* All authors contributed in devising the study. RB processed NEMO and CMIP6 model data and SR undertook the remaining data processing, data analysis, and lead manuscript writing. All authors contributed to interpretation of the results and reviewing the manuscript.

*Competing interests.* The authors declare no competing interests.

*Acknowledgements.* The authors are very grateful to Dr. Julia Pfeffer and two anonymous reviewers for their comments and constructive
criticisms of the Discussions manuscript.

The authors were all supported by European Research Council (ERC) under the European Union's Horizon 2020 research and innovation
programme under grant agreement No 694188, the GlobalMass project (globalmass.eu). JLB was additionally supported through a Lever-
hulme Trust Fellowship (RF-2016-718) and a Royal Society Wolfson Research Merit Award. We would like to thank Richard Westaway
(University of Bristol) for project management and in editorial of the final manuscript for language and publication quality review.
The authors are grateful for the open availability of observational and derived data sets, as referenced in the text and data availability
section.

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
