# Peer review of "Attributing decadal climate variability in coastal sea-level trends"

_Ocean Science, 2022_

## Author Comment (AC1)

Dear anonymous reviewer,

The authors thank the reviewer for their comprehensive review. We provide the following comments (in blue, italic) to the reviewer's major concerns (in black) below. Minor comments, typographical and figure errors are amended in the revised manuscript.

The study "Attributing decadal climate variability in coastal sea-level trends" aims to relate the variability in decadal sea-level trends in coastal regions to climate variability. Based on a high resolution ocean model, the authors reconstruct coastal, sea-level trends via linear relationships with climate indices.

Based on their reconstruction, the authors can confirm and quantify the dominance of manometric over steric sea-level trends at coastal locations and pinpoint locations where the GRD signal is of importance. They quantify the sea-level trend variance explained by climate variability and show that in one-third of all coastal locations almost half the variance can be explained by climate variability. Finally, their results suggest that climate variability has suppressed sea-level rise during the period 2008-2018.

The results are well presented, the paper is relevant to the scientific community and doesn't present any major flaws. I recommend the article to be accepted, after some minor revisions.

**We thank the reviewer for their precis and recommendation.**

**General comments**

1. The authors used a series of climate indices to establish the relation between sea-level trends and climate variability. It is my understanding that, except for the AMOC index, the indices are based on observations, or reanalysis data. Wouldn't it be more consistent to use indices that are derived from the model output or the atmospheric forcing data set to infer the relation between those indices and the modeled sea level? The reconstructed sea-level trends that are compared to observed trends should still be based on the observed indices, of course.

In terms of consistency, yes we agree that it is more consistent to use the model atmospheric forcing and model ocean variables to derive the climate indices and their decadal trends. In doing so, we should expect the relationship (regression coefficient) between the climate indices and sea-level trends to be stronger, in particular since the steric sea level change is directly related to the model sea surface temperature (SST). However, one aim of the work is to demonstrate the applicability of the derived relationship between sea-level trends and climate indices with observations. So in mind of the practicality of stakeholders using the relationship derived here, we felt it was more representative to use standard, publicly available derivation of the climate mode indices (CI). Although there are minor differences in the model atmospheric forcing and model SST compared with the standard CI, the calculated decadal trends are similar (Fig R1.1). Most of the acceleration or deceleration of the CI trends is captured in the NEMO model or its atmospheric forcing. The modes of variability, now discussed more in the revised manuscript, are dominated by relationships with ENSO, PDO, AMOC, with second or third order PCs additionally correlated with SAM or AO, and IOD with lower-order PCs. Whilst there are some differences such as the ENSO trends in the late 1970's, the SAM index in the early 1990's and the IOD index before 1985, that could affect the exact value of the coefficients determined by this method, the dominant modes of variability and the summed reconstructions would not be significantly affected if we were to use the NEMO-derived climate indices rather than the standard. Therefore, the conclusions of the work aren't affected.

Fig R1.1 Comparison of decadal trends in climate mode indices (CI) by publicly-available standard derivation (Solid lines) and recalculated from the NEMO model atmospheric forcing and modelled SST (dashed lines)

2. The authors limit their analysis to a coastal region within 25 km of the coastline. As they point out the translation of steric to manometric sea-level anomalies depends on the water depth. I am wondering if a criterion based on water depth to identify coastal region would be more appropriate?

The authors did evaluate three different ways to define 'coastal':

- 1. the standard altimetry community's definition of bathymetry less than 125 m, which represents 4.5% of the NEMO grid points;
- 2. the shallow bathymetry definition of Penduff et al (2019) of less than 25 m depth, which represents 1.3% of the NEMO grid points; and,
- 3. the Penduff et al (2019) distance definition of within 25 km of any low (coarse) resolution GHSSH coastal polyline, which covers 2.5% of the NEMO grid points.

Option 2, the shallow bathymetry definition of Penduff et al (2019) didn't provide adequate coverage of coastal locations across the globe in the NEMO grid so we neglected it. The option 3 distance definition gave a good coverage of valid grid locations in the NEMO grid. The additional grid points covered by the option 1 altimetry community definition of shallow water (125 m) gave additional coverage away from the coastline itself and due to the nature of sea-level variability, it did not provide additional information in the principal component analysis of the sea-level trends. The authors felt the distance from coastline definition is sufficiently representative of the coastal region, given we use a principal component analysis to characterise the sea-level trend variability. Fig R1.2 presents the SSH leading EOF in each basin, based on options 1 (a) and 3 (b), demonstrating the limited difference the choice makes to the analysis against CI.

---

## Author Comment (AC2)

**Response to reviewer 2 comment,: RC2, on "Attributing decadal climate variability in coastal sea-level trends" by Royston, Bingham and Bamber**

Dear anonymous reviewer,

The authors thank the reviewer for their review. We provide the following comments (in blue, italic) to the reviewer's major concerns (in black) below.
* * *
The authors showed the effect of climate modes on the decadal sea level rise in the coast using the sea level modeling results of the high-resolution ocean model and the CMIP6 models. In general, the ideas are clear and scientifically supported. However, some corrections are suggested to aid the reader's understanding.

- Why don't the authors show the mean value of decadal trends in Figure 1 or 2?

*Our work aims to quantify the variability about the time-mean, regional trend that is driven by climate variability on decadal time scales. The time-mean regional trends are derivable from observations and are discussed in detail in CMIP modelling work, for example (e.g. Meyssignac et al 2017). For transparency and completeness, the time-mean regional trends derived in the NEMO model run are added to the Supplementary Information, Fig S2.*

- In Figure 1, there is not enough explanation for each panel. Adding more information about each picture to the picture caption is suggested.

*The Figure shows the standard deviation of the rolling decadal trends from the NEMO model and the ensemble mean and spread of these standard deviations. The text has been modified in the revised manuscript.*

- In Figure 2, since opaque rectangles are overlapped, information distortion is possible. Instead, it is recommended to minimize the overlap by averaging several boxes.

*The authors appreciate the feedback on this presentation issue. By averaging the overlapping grid node values we would smooth the result, which isn't ideal. With only 1 month to respond to comments we haven't made a change to this Figure; if deemed necessary by the Editor then please allow more time to modify the Figures.*

- It is proposed to add the total sea level rise rate to Figure 2. If the sea level rise rate is very low, this classification may not have much meaning.

*The aim of the paper is to investigate the variability in decadal sea-level trends about the time-mean and spatially about the global-mean. This Figure shows the variance in the decadal trends explained by different sea-level components, as proportions of the variability in the rolling decadal trends (the sum of the time-mean trend and variability shown in Fig 1a). The Figure caption has been amended to hopefully make this clearer.*

- In Figure 3, it is proposed to verify and show the results of reconstruction and NEMO for Tide Gauge. The authors did not show their level of accuracy.

*The correlation and variance explained from the reconstructions for the tide gauge locations given in Fig. 4 are presented within triangles with white borders overlaid on the coastal grid points in Fig. 3. We have not shown the accuracy of the NEMO model to replicate the tide gauge observations, but the NEMO citations and our Supplementary Information does show NEMO replicates the open ocean*

**Response to reviewer 2 comment,: RC2, on "Attributing decadal climate variability in coastal sea-level trends" by Royston, Bingham and Bamber**

*observations well on the decadal timescale. Our discussion for tide gauges only aims to demonstrate the potential for reducing observations by climate-driven variability.*

---

## Author Comment (AC3)

**Response to comments by reviewer 3, Dr. Julia Pfeffer: RC3, on "Attributing decadal climate variability in coastal sea-level trends" by Royston, Bingham and Bamber**

Dear Dr Pfeffer

The authors would like to thank you for your comprehensive review and very helpful, constructive suggestions that will improve the paper. We provide the following comments (in blue, italic) to the reviewer's general and detailed comments (in black) below.

**Summary**: The manuscript by Royston et al., explores the mechanisms responsible for sea level variability at decadal time-scales using a combination of ocean general circulation model (NEMO), climate model predictions (CMIP6 ensemble), satellite altimetry observations and tide gauge observations. The authors attempt a reconstruction of sea level trend anomalies using a regression between climate indices and the correlated components of a decomposition in empirical orthogonal functions of the NEMO sea level outputs. These reconstructions are compared with tide gauge and satellite altimetry observations, to estimate how climate modes contribute to decadal sea level variations.

**Recommendation:** The thematic treatment is of great interest for the scientific community, as the internal sea level variability has been identified as a major source of uncertainties in climate models, especially in the near-future. It is therefore important to advance knowledge in the identification of the mechanisms responsible for sea level variations in a changing climate. The study brings some useful insights in this regard, and should be considered for publication after consideration of the following comments.

*We thank you for your precis, recommendation and very constructive comments.*

**General comments:**

The description of the method lacks clarity in the manuscript. It is difficult to follow step by step what has been done with which data. It is unclear to me how the CMIP6 predictions are used. The reconstruction seems to be applied on the manometric, steric and GRD outputs of the NEMO predictions, but the method is still unclear. Have the eof decompositions been applied on the total, manometric, steric and GRD contributions individually? Then correlations are estimated between the PCs of the eof decompositions and the climate indices. Finally, a regression analysis is performed, though it is unclear how. A few equations would help to better understand this final stage. The text should be clarified and a flow chart would help to picture the steps of the analysis.

*We have added a paragraph at the beginning of the Method section to summarise the method, and then follow with detail. Yes, the EOF decomposition, to reduce the size of the problem by dimension reduction in space, is applied to each sea-level component and each ocean basin separately. To reconstruct the decadal sea-level variability for each component part from just climate mode indices, we associate each climate index with one PC, and use a linear regression to project the climate mode index on to each PC. We reconstruct the sea-level components from the sum of reconstructed PCs and total SSH from the sum of component parts. This way, each climate index is associated with a sea-level component EOF; we use the maximal correlation coefficient between climate indices and PC to pick a 'most appropriate' climate index. The choice of method is discussed more below.*

There are a few methodological hindrances in the approach of the authors that have not been acknowledged. In particular, the authors calculate the correlations with climate indices based on the

**Response to comments by reviewer 3, Dr. Julia Pfeffer: RC3, on "Attributing decadal climate variability in coastal sea-level trends" by Royston, Bingham and Bamber**

results of an eof decomposition. The eof decomposition will pull apart physical signals and redistribute them into statistical modes explaining less and less variance as you increase the order. As a consequence, the correlation between sea level changes and individual climate indices might be lost because it has been divided into several modes of variation. To avoid this issue, a multivariate regression is usually carried out directly on the variable of interest (here sea level changes). To deal with the issue of intercorrelated climate indices, a regularisation can be applied (see Pfeffer et al., 2018 and 2022). The multivariate regression also allows the identification of climate indices contributing to the sea level variations at each grid point, which is only possible with limitations with the author's approach. The authors should acknowledge these limitations to allow the reader to assess the relevance of the approach.

*Thanks for the constructive feedback. There are two issues to discuss here, the EOF approach for the sea-level trends and the choice of regression to relate the sea-level trend variability to climate variability.*

*We apply the EOF analysis to coastal sea-level trends to reduce the spatial dimensions, because there is a lot of redundancy applying a regression at each grid point (because of the spatial covariance of sea level). This approach reduces the size of the data set provided, and reduces the computational burden to reconstruct sea-level variability from new climate data across the global-coastal locations we consider compared with reconstructing at each grid point. (The data set comprises 3-7 EOF patterns with one set of regression coefficients per sea-level component and basin; rather than a different set of regression coefficients for each grid point). However, we do agree that the EOF analysis, by its orthogonality, can separate signals of a given physical driver into multiple EOF modes. A consequence of this is, the variance of our climate-driven reconstructed sea-level trends will be a lower limit on the true variance that relates to climate variability. We have added a paragraph to the Methods section to highlight this to the reader.*

*To better describe and discuss how we project the climate index variability on to the PCs of sea-level trend variability, we have added a paragraph to the Methods section. We agree that the multi-variate approach with a ridge regression does limit the impact of multicollinearity on inflated regression coefficients. When a problem has multicollinearity in the explanatory variables and auto-correlation in both explanatory and target variables, the analytical least squares solution is unobtainable. As Pfeffer et al (2018; 2022) discuss, for a MVLR with a ridge regression or regularisation, there is a parameter that needs tuning in the penalty term, usually done through cross-validation. Our approach, by only regressing one climate index against one PC at a time (orthogonal from all other PCs) and summing up, ensures the projected reconstruction isn't inflated due to multicollinearity, i.e. the same driving mechanisms isn't duplicated in the reconstruction. But as a consequence of the approach we use, it is certainly plausible that climate-variability-driven sea-level variability will be split between EOF modes and not fully replicated in our reconstruction, leading to a lower sea-level variance in the reconstruction.*

*We have modified the Methods paragraph Lines 113-116 in the original submission to add a discussion of the consequence of these combined method choices.*

The description of the data lacks clarity in the manuscript. In particular the description of processing applied on the altimetry and tide gauge measurements is imprecise. It is not clear that adequate corrections have been applied for the various datasets for GIA and GRD.

*The paragraph 3.4 and 3.6 are amended to provide clarity. We have added explanation of the correction of ESA SLCCI v2 gridded data for GIA and GRD and for the tide gauge data we explain we don't apply a GRD correction, but simply remove the mean trend in the Figures. We note, the primary purpose of this work is to quantify the temporal variability of sea-level trends, spatially or regionally,*

**Response to comments by reviewer 3, Dr. Julia Pfeffer: RC3, on "Attributing decadal climate variability in coastal sea-level trends" by Royston, Bingham and Bamber**

*and reconstruct the variability associated with climate variability. In terms of both the global GIA and global GRD corrections applied to the satellite altimetry, these only shift the trends in Fig 5c. Similarly for the tide gauge trends, the time-mean trend or trend corrections for GRD or GIA would only shift the centre in Fig 4. The spatial redistribution due to GRD and spatial GIA are very small and don't materially affect the conclusions drawn in the work. These corrections are applied for completeness.*

The description of the results is clear and interesting. However, more precision would be appreciated. In particular, the authors restitute the performance of the sea level reconstruction based on climate modes by reporting the percentage of variance explained above a certain threshold. It would be much more informative to have a range of variance, with a minimum and maximum bounds for a given region. The authors also use several time expressions like "explain much of this" or "explain well", it would be useful to have a metric, so that the reader can assess what "much" or "well" means.

*The text throughout the manuscript has been amended to quantify coverage or variance explained, rather than using "much". We use correlation and variance explained at each location to quantify the climate-associated decadal sea-level variability against the NEMO model and validate against observations. The authors feel that the variance explained metric at each location (Fig 3) gives a fairer impression, with spatial distribution, of how much of the decadal sea-level variance is explainable by climate variability, compared with deriving basin-average percentiles and max-min bounds. We have added 25% and 75% percentiles of variance explained in Table 1.*

The conclusion is clear, but fails to compare the results with Pfeffer et al., (2018) and (2022) dealing with the attribution of climate modes contributions to steric and manometric sea level changes.

*We have added discussion of these papers to the Introduction and Methods section, and we have added comparison with previous literature on the derived leading modes of climate variability to the Results and Conclusions sections. Considering this comment and comments in RC1, we have added a Table 3 and text in Sections 4.4 and 5 to compare with previous works.*

**Detailed comments:**

Abstract: Define GRD or use full words *Amended*

L28-29: formulation not excessively clear *Amended the sentence structure, and moved to next paragraph, with Line 33 comment.*

L33: "A proportion of regional variation in sea level rise": change rather than rise. The full sentence is not clear. *Amended sentence structure.*

L44: The two following references are lacking. Pfeffer et al., 2018 has shown the influence of the PDO, ENSO, AMO, NPGO and IOBM on steric sea level changes, with significant influence at pluri-decadal time scales. Pfeffer et al., (2022) has shown the influence of NSO, PDO, AO, NAO and SAM modes in the barystatic component of sea level measured by GRACE. *We thank the reviewer for bringing these papers to our attention and we add citations and discussion in the Introduction and Discussion.*

L70: sentence not clear *Amended*

L70-72: see general comment on eof decomposition *Added text to Introduction and Method*

**Response to comments by reviewer 3, Dr. Julia Pfeffer: RC3, on "Attributing decadal climate variability in coastal sea-level trends" by Royston, Bingham and Bamber**

L78-81: reformulate to increase clarity *Amended*

L102: not a huge fan of rolling pin, that will generate an aliasing of many different signals and modes of variability *We agree that taking a rolling procedure in time will cause some aliasing. However, the same approach is applied to both the target and explanatory variables, so the same aliasing occurs in both. Ideally, longer time series would be investigated with a better filter, but with only 58 years of high-resolution model data, the rolling approach is a compromise to obtain a useful length of data.*

L117: why not using the full altimetry period? *We have amended the text to be clear, this is just one test to validate the method. The same could be applied to the whole altimetry period.*

L110-112: verb missing. Reformulate the sentence for clarity *This paragraph has been amended to better explain the method and discuss its limitations. Hopefully now clearer.*

L113-115: Not clear reformulate. *This paragraph has been amended to better explain the method and discuss its limitations. Hopefully now clearer.*

L123-124: not clear why GRD correction is not applied. It does not rely on GPS measurements. *Yes, the theoretical GRD from a sea-level fingerprint approach could be corrected in the tide gauge measurements. There would remain other sources of VLM in the tide gauge data. GRD is predominantly linear trend in time and for much of the tide gauge network the effect is quite small in magnitude, though we acknowledge for some sites the influence is significant (our Fig 2). The primary aim of using tide gauge data was to validate the reconstruction of variability associated with climate variability, about the mean trend. Including a GRD correction would have negligible impact on the conclusions drawn, but we acknowledge in future work including the non-linear correction should be done.*

L133:typo "noting" *sentence amended following Reviewer 1 comments.*

L157-158: This sentence is very confusing. GRD and GIA are observed by satellite radar altimetry and by tide gauges, but not in the same way since tide gauges are attached to the coast. The corrections applied on the various datasets must be consistent one with another. If you wish to remove these effects from altimetry, you need to remove the global mean correction if it has been applied if it has been applied (it depends on the product chosen, but usually gridded altimetry products are not corrected for GIA), and then apply an appropriate correction at each grid point. Maybe consider writing this paragraph after the description of the datasets. So it would be easier for the reader to understand what data processing is applied to which data. *Wording amended and section moved.*

L169: "Absolute sea level is defined from a multi-mission" → Absolute sea level is defined from **the ESA SLCCI v2** multi-mission *Amended*

L171-173: it is not clear that appropriate correction has been applied for GIA. As stated earlier, altimetry-based gridded SLA products do not usually (check specific product) correct for GIA. The GIA correction is usually only applied on the GMSL. Please reference the altimetry product in greater detail (exact product name, version and DOI) and explain what GIA correction has been applied in it. Then, state what specific correction you applied, so that it is consistent with other datasets. *This paragraph has been re-written, hopefully it is clearer. The ESA (2018) citation gives the DOI for the ESA SLCCI v2 product. We apply the ICE6G GIA correction that includes the global-mean trend and the spatial component for the geoid, as derived from spherical harmonic coefficients given by Peltier.*

L186: This approach has flaws. An EOF decomposition will pull apart the physical signal into a suite of statistical modes. As a consequence, coherent physical signals will be separated into several modes. If

**Response to comments by reviewer 3, Dr. Julia Pfeffer: RC3, on "Attributing decadal climate variability in coastal sea-level trends" by Royston, Bingham and Bamber**

the sea level is influenced by one or several climate modes at one location, the part of variance explained by climate indices is likely to be separated into several modes as well. Therefore, you will not be able to retrieve a strong correlation with a single PC, but are more likely to get partial correlations with a lot of different PCs. This is why multivariate regression is preferred. To deal with the issue of correlated indices a regularisation might be applied. Alternatively, statistical tests have also been applied to determine the robustness of a correlation between two time series. *Additional text has been added to the Method section to discuss the limitations of the approach and alternatives, as per the Major Point 2. The original manuscript discussed some of the limitations in Section 4.4 and we have added text here specifically relating to this concern.*

L192: Why are tide gauges not corrected for GRD? Admittedly there are other sources of deformation that cannot be easily modelled and require GPS observations that are very sparse and usually very limited in time, but non-linear GRD effects can be estimated with models. *See L123-124 comment.*

L200: it would be interesting to see the differences between the NEMO run and the CMIP6 mean prediction. It would be easier to compare to the spread, in order to assess if both approaches are consistent within uncertainties. *Figure 1 has been amended to include this comparison.*

Section 4.1: clear and interesting

L231-246: The results might be inflated to some extent in this section. The proportions of variance explained are credible and exhibit similar order of magnitudes than previous studies. It is perfectly fine to report an explained variance of 20 or 30%. It is still significant when compared with the accuracy of model and observations, but also with other physical signals present in sea level observations, predictions and reconstructions. It would probably be better to give a range of explained variance for a given region, rather than a minimum explained variance. The regions where the percentage of explained variance is small (~ <30%) cover most of the coastal areas of the world (orange areas in Fig. 3b). It is important to state that in most coastal areas of the world climate modes explain a small but significant part of the variance. Similarly in Table 1, it would be better to report the percentage of locations with a variance in the first (0-25%), second (25-50%), third (50-75%) and fourth (75-100%) quartiles. That way, the reader would have a better picture of the statistical distribution of the results. *Text has been amended to highlight many regions the reconstruction does not explain much decadal-scale variance. We have added 25/75%iles to Table 1, in addition to 33/50/67%.*

L265: name the regions where coastally trapped wave are expected *Amended. On these time scales, it is probably more correct to*

L272: Some precisions would help here. Which region are you referring to? What constitutes large magnitude variability? Is it above a certain threshold of RMS? Which one? What constitutes "much of that decadal signal" (proportion?)? *Amended.*

L278-279: the reconstructed trend anomaly seems to capture the pattern well but not the amplitude. It should be said. A figure of the difference would help. For regions where the observed trends anomalies are large (e;g. tropical Pacific, west coast of North and South America etc.) it would be good to estimate the ratio between the reconstructed and observed trend anomaly. Also be careful about the vocabulary, it is a trend anomaly not a trend. *As discussed elsewhere, the method by using EOF analysis and a single explanatory climate index per PC, may give a lower estimate of the variance associated with climate variability. 'Trend anomaly' corrected in the revised manuscript.*

L311-313: this has also been shown by Pfeffer et al., 2018 for the steric component, with in particular the influence of AMO emerging in ocean reanalyses, only with a sufficient time coverage (~ 50 years).

**Response to comments by reviewer 3, Dr. Julia Pfeffer: RC3, on "Attributing decadal climate variability in coastal sea-level trends" by Royston, Bingham and Bamber**

Other modes such as ENSO, PDO, NPGO (North Pacific Gyre Oscillation), IOD and IOBM (Indian Ocean Basin Mode) have been shown to have a strong influence on the interannual-variability of steric sea levels over a 57 time period. The NPGO is not often considered, though it has been shown to have a very large influence on SSH (see articles by Di Lorenzo including but not limited to Di Lorenzo et al., 2008) and on the manometric component (Pfeffer et al., 2022). *The NPGO is sometimes derived from the second mode of SSH anomalies, rather than the second mode of SST. So it can be derived from the same data as we are attempting to replicate or understand. I appreciate that the other climate indices based on SST (ENSO, PDO, IOD mode index) also clearly relate to steric sea level, though indirectly. Citation added to the revised manuscript and more discussion has been made of previous literature in the Introduction, Results and Conclusions.*

Section 5 Conclusion: please provide metrics in your conclusion to support the soundness of your approach *We have added some quantified statements to the conclusions.*

**References:**

Pfeffer, J., Tregoning, P., Purcell, A., & Sambridge, M. (2018). Multitechnique assessment of the interannual to multidecadal variability in steric sea levels: A comparative analysis of climate mode fingerprints. *Journal of Climate*, *31*(18), 7583-7597. https://doi.org/10.1175/JCLI-D-17-0679.1

Pfeffer, J., Cazenave, A., & Barnoud, A. (2021). Analysis of the interannual variability in satellite gravity solutions: detection of climate modes fingerprints in water mass displacements across continents and oceans. *Climate Dynamics*, 1-20. https://doi.org/10.1007/s00382-021-05953-z

Di Lorenzo, E., Schneider, N., Cobb, K. M., Franks, P. J. S., Chhak, K., Miller, A. J., & Rivière, P. (2008). North Pacific Gyre Oscillation links ocean climate and ecosystem change. *Geophysical Research Letters*, *35*(8).